# Molecular Classification and Tumor Microenvironment Characterization of Gallbladder Cancer by Comprehensive Genomic and Transcriptomic Analysis

**DOI:** 10.3390/cancers13040733

**Published:** 2021-02-10

**Authors:** Nobutaka Ebata, Masashi Fujita, Shota Sasagawa, Kazuhiro Maejima, Yuki Okawa, Yutaka Hatanaka, Tomoko Mitsuhashi, Ayako Oosawa-Tatsuguchi, Hiroko Tanaka, Satoru Miyano, Toru Nakamura, Satoshi Hirano, Hidewaki Nakagawa

**Affiliations:** 1Laboratory for Cancer Genomics, RIKEN Center for Integrative Medical Sciences, Yokohama 230-0045, Japan; nobutaka.ebata@riken.jp (N.E.); m-fujita@riken.jp (M.F.); shota.sasagawa@riken.jp (S.S.); kazuhiro.maejima@riken.jp (K.M.); yuki.okawa@riken.jp (Y.O.); ayako.oosawa@riken.jp (A.O.-T.); 2Department of Gastroenterological Surgery II, Faculty of Medicine, Hokkaido University, Sapporo 060-8638, Japan; torunakamura@med.hokudai.ac.jp; 3Research Division of Companion Diagnostics, Hokkaido University Hospital, Sapporo 060-8638, Japan; yhatanaka@huhp.hokudai.ac.jp; 4Department of Surgical Pathology, Hokkaido University Hospital, Hokkaido 060-8648, Japan; mitsut74@huhp.hokudai.ac.jp; 5Laboratory of DNA Information Analysis, Human Genome Center, The Institute of Medical Science, The University of Tokyo, Tokyo 108-8639, Japan; hiroko@hgc.jp (H.T.); miyano@hgc.jp (S.M.)

**Keywords:** gallbladder cancer, tumor microenvironment, EMT, TGF-β signaling pathway

## Abstract

**Simple Summary:**

Gallbladder cancer (GBC) is a rare but lethal cancer. Molecular characterization of GBC is insufficient so far, and a comprehensive molecular portrait is warranted to uncover new targets and classify GBC. Clustering analysis of RNA expression revealed two subclasses of 36 GBCs, which reflects the status of the tumor microenvironment (TME) and poor prognosis of GBC, including epithelial–mesenchymal transition (EMT), immune suppression, and the TGF-β signaling pathway. The knockout of *miR125B1* in GBC cell lines decreased its invasion ability and altered the EMT pathway. Mutations of the genes related to the TGF-β signaling pathway were enriched in the poor-prognosis/TME-rich cluster of GBCs. This comprehensive molecular analysis provides a new classification of GBCs based on the TME activity, which is involved with EMT and immune suppression for poor prognosis of GBC. This information may be useful for GBC prognosis and therapeutic decision-making.

**Abstract:**

Gallbladder cancer (GBC), a rare but lethal disease, is often diagnosed at advanced stages. So far, molecular characterization of GBC is insufficient, and a comprehensive molecular portrait is warranted to uncover new targets and classify GBC. We performed a transcriptome analysis of both coding and non-coding RNAs from 36 GBC fresh-frozen samples. The results were integrated with those of comprehensive mutation profiling based on whole-genome or exome sequencing. The clustering analysis of RNA-seq data facilitated the classification of GBCs into two subclasses, characterized by high or low expression levels of TME (tumor microenvironment) genes. A correlation was observed between gene expression and pathological immunostaining. TME-rich tumors showed significantly poor prognosis and higher recurrence rate than TME-poor tumors. TME-rich tumors showed overexpression of genes involved in epithelial-to-mesenchymal transition (EMT) and inflammation or immune suppression, which was validated by immunostaining. One non-coding RNA, *miR125B1*, exhibited elevated expression in stroma-rich tumors, and *miR125B1* knockout in GBC cell lines decreased its invasion ability and altered the EMT pathway. Mutation profiles revealed *TP53* (47%) as the most commonly mutated gene, followed by *ELF3* (13%) and *ARID1A* (11%). Mutations of *ARID1A*, *ERBB3*, and the genes related to the TGF-β signaling pathway were enriched in TME-rich tumors. This comprehensive analysis demonstrated that TME, EMT, and TGF-β pathway alterations are the main drivers of GBC and provides a new classification of GBCs that may be useful for therapeutic decision-making.

## 1. Introduction

Gallbladder cancer (GBC) is a rare tumor that exhibits some regional differences worldwide. The highest incidence of GBC is reported in South America and Asia including Japan, wherein the GBC incidence rate is 7/100,000 [1]. In the USA, the incidence rate of GBC is only 1.6/100,000, and racial differences in GBC etiology is large, although gallbladder stone or chronic inflammation is one of the main risk factors of GBC [1]. GBC is a lethal cancer with an overall mean survival rate of 6 months and a 5-year survival rate of 5% [2], primarily owing to non-specificity of symptoms during initial stages of the disease and late diagnosis at advanced or non-curative stages. GBC has the worst survival among diverse biliary tract cancers (BTCs), even after chemotherapy [3]. Surgical resection is the best regimen that increases the chances of long-time survival in patients with localized GBCs, but it is difficult to curatively resect advanced GBC owing to its anatomical location and highly locally invasive or metastatic potential. Gemcitabine, the first-line approved treatment for locally advanced or metastatic GBC, exhibits very limited effects [4], and no molecular-based therapies are yet approved for GBC.

Molecular profiling is critical to establish treatment strategies for lethal cancers, and molecular characterization of GBCs has been previously attempted. A study [5] revealed that genomic alterations frequently occurred in *TP53* (47.1%), *ERBB3* (11.8%), and *KRAS* (7.8%) in Chinese GBCs, while another study observed *TP53* (57.9%), *CDKN2A/B* (25%), and *SMAD4* (17%) mutations in GBCs in the US [6]. A recent study observed recurrent alterations in *ELF3* and the WNT pathway in Indian and Korean GBCs [7]. Among these driver mutations, *TP53* mutations play a central role in GB carcinogenesis, while *KRAS* mutations are the most dominant in other types of BTCs [8]. RNA expression analysis of the coding genes in BTCs revealed the upregulation in the expression of the immune-related transcriptome in GBCs and showed no significant difference in disease-free survival according to tumor locations [8]. Peng et al. reported the members of the miRNA-200 family that were downregulated in BTC samples [9] and known to participate in the epithelial–mesenchymal transition (EMT) process by directly targeting E-cadherin repressors [10]. The long non-coding RNA (lncRNA) *MEG8* contributed to the progression of EMT in lung and pancreatic cancers [11], while *MEG3* expression was downregulated in GBCs and correlated with poor prognostic outcomes [12]. Considering the heterogeneity of GBCs and BTCs, the current molecular characterization and known molecular markers are insufficient, necessitating further evaluation of the genome and RNA expression data of GBCs to improve outcomes of affected patients.

To further characterize the molecular features of GBCs and compare them with other types of BTC, here we performed a comprehensive analysis of the genome and expression of RNAs, including ncRNAs, in 36 GBCs and 8 BTCs of the hilar bile duct (HBDC). Although the transcriptomic difference based on tumor location in the bile duct tree was unclear, we could classify GBCs into two subclasses based on the tumor microenvironment (TME) and patient prognosis. We also identified the alterations in the expression of some ncRNAs and genes related to TME, such as those related to EMT, tumor immunology, and the transforming growth factor (TGF)-β signaling pathway. This comprehensive analysis may elucidate GB carcinogenesis and provide a new classification of GBCs that may be useful for therapeutic decision-making.

## 2. Results

### 2.1. RNA-Sequencing (RNA-Seq) and Clustering Analysis Revealed Two Subclasses of GBCs

It is sometimes difficult to discriminate between the anatomical location of GBC and a tumor developed in the upper common bile duct or hepatic bile duct, even after pathological examination following resection [13]. First, to explore the molecular differences between gallbladder body and neck cancer (GBBC) and cystic duct cancer (CDC), which are anatomically defined as GBCs, and hilar bile duct cancer (HBDC), we performed RNA-seq of frozen surgical specimens of 20 GBBCs, 16 CDCs, and 8 HBDCs located very closed to the junction of cystic and common hepatic ducts, as evaluated by pathological examination of resected samples. We extracted total RNAs from these 44 frozen surgical specimens and constructed RNA-seq libraries after rRNA depletion, which included ncRNAs and non-polyA-tailed RNAs. We performed hierarchical clustering based on gene expression detected in these tissues by RNA-seq, and only genes with high expression (fragments per kilobase of exon per million fragments mapped (FPKM) > 2 in more than 12 samples) were used for clustering. The clustering analysis of RNA-seq data revealed no apparent difference between HBDCs, CDCs, and GBBCs (Appendix A), indicating their similar molecular features.

We then focused on GBBCs and CDCs (these two cancers are anatomically GBCs) for clustering and examined genomic differences that may contribute to poor prognosis. Clinical features of patients with GBC are shown in Appendix A. Patients with GBC in this study showed significant difference in prognosis based on their disease stage (T and N factors) but not anatomical locations (CDC or GBBC) (Appendix A). Hierarchical clustering based on the expression of genes, including ncRNAs, allowed classification of GBCs into two clusters, namely Cluster A and Cluster B (Figure 1a). Cluster B showed significantly poorer prognosis and higher recurrence rate (*p* = 0.01 and 0.005 by log-rank test, respectively) with HR: 6.6439, 95% CI (1.514–29.16) and HR: 3.623, 95% CI (1.226–10.71) than Cluster A (Figure 1b). However, the difference of gender, CDC/GBBC, pT (T1+T2 vs. T3+T4), and pN+ between Clusters A and B did not reach statistical significance (*p* > 0.05 by Fisher’s exact test, Appendix A). The Cox proportional hazard regression analysis also showed that the overall survival (OS) depended on pT factor but that the disease-free survival (DFS) depended on pN factor and Cluster A/B (Appendix A), suggesting that Cluster A/B could be an independent factor associated with GBC prognosis.

### 2.2. The High-Stromal Cluster Showed Upregulated Expression of Genes Involved in EMT and Inflammation

To characterize the expression of the genes of these clusters, we performed gene set enrichment analysis (GSEA) between Clusters A and B using the Molecular Signatures Database (MSigDB) as the reference gene set. We found 19 gene sets that were differentially expressed (false discovery rate (FDR) < 0.25) between Clusters A and B (Appendix A). Figure 2a shows the list of the gene sets that were upregulated in Cluster A (right) or B (left) in GSEA based on the normalized enrichment score (NES). In particular, the gene sets related to EMT and TGF-β signaling were significantly upregulated in Cluster B (*q*-values = 0.001582, and 0.028345, respectively, Figure 2b). EMT is involved in cancer cell invasion and progression and can be associated with poor prognosis in Cluster B (Appendix A). The TGF-β signaling pathway also upregulates the expression of EMT-related genes and plays a central role in EMT and TME [14]. A broad spectrum of immunological pathways, including inflammatory response, interferon-γ and -α responses, and TNF-α signaling, was also upregulated in Cluster B, while fatty acid and bile acid metabolisms were enriched in Cluster A (Figure 2a right). These observations suggest that the function of normal gallbladder or biliary tract was retained in GBCs from Cluster A.

Thus, TME may play a critical role in GBCs in Cluster B. To further investigate the TME of GBCs, we analyzed these expression profiles by “Estimate”, which can estimate stromal cells and the level of immune cell infiltration in tumor tissues based on expression data, as well as tumor purity (Appendix A). The Estimate analysis also showed that immune and stroma scores were significantly higher at Cluster B than at Cluster B (*p* < 0.01 by Mann–Whitney U test, Figure 2c).

### 2.3. The High-Immune Core Cluster Showed Increased Expression of Immune Checkpoints

We next examined the differences in the expression of individual genes between Clusters A and B. The comparison of expression profiles revealed 3997 protein-coding genes and 1910 non-coding genes that were differentially expressed between the two clusters (Appendix A). The expression levels of known regulators of EMT, such as *DST*, *SNAI1*, and *SNAI2*, are shown in Figure 3a. We also evaluated the expression of immune genes and those encoding immune check-point molecules such as *PD-1*, *PD-L1*, *CD8A*, *CTLA4*, *LAG3*, *TIGIT*, *TIM-3, PD-L2*, *IFNG*, *CD4*, *CD45*, *CD3E*, *CD163*, *TGFB1*, *IL10*, and *FOXP3*, and found that these genes were significantly upregulated in Cluster B (*p* < 0.02 by Mann–Whitney U test, Figure 4a and Appendix A). This result suggests that GBCs in Cluster B are immunologically ‘hot’ tumors. However, many immunosuppressive cell markers such as *FOXP3* (Treg) and *CD163* (M2 macrophage) and immunosuppressive cytokines such as *IL-10* and *TGFB1* were also upregulated in Cluster B, indicating the complicated immune microenvironment of GBC tissues [15,16].

### 2.4. Immunostaining of Stromal Markers and Immune Cells in GBCs

To validate these stromal and immune reaction markers of GBCs, we performed immunostaining for vimentin, CD8, and T-cell immunoglobulin mucin-3 (TIM-3) on 35 formalin-fixed paraffin-embedded (FFPE) GBC slides obtained from the same patients and compared the expression of markers between Clusters A and B. Vimentin is a marker of mesenchymal-derived cells and cancer-associated fibroblasts (CAFs), which involve a cancer stromal component and TME [17]. Immunohistochemical staining for vimentin showed positive staining for stromal cells, not tumor cells, and significantly higher vimentin expression in GBCs of Cluster B than in GBCs of Cluster A (*p* = 0.0003308 by Mann–Whitney U test, Figure 3b,c). This observation correlated well with the calculated stromal score by RNA-seq. 

CD8 is a maker of cytotoxic T lymphocytes [18], while TIM-3 indirectly suppresses effector T-cell activity by acting on myeloid-derived suppressor cells, Tregs, and dendritic cells [19,20]. We counted the number of positively stained cells at “hot spots” and found more positive cells near the tumor cells. We also confirmed that Cluster B samples had more CD8- and TIM-3-positive cells than Cluster A samples (*p* = 0.03876 and 0.02766 by Mann–Whitney U test, respectively) (Figure 4b,c), consistent with the computational estimates of RNA expression results and immune scores. Many of these positive immune cells were present in the stroma close to the tumor cells and were often solidified (Figure 4d for CD8 and Figure 4e for TIM-3).

### 2.5. Non-Coding RNAs Are Involved in the Invasion and EMT of GBCs

We sequenced both coding and ncRNAs of GBCs after rRNA depletion (See Methods) and found 437 ncRNAs that were significantly differentially expressed between Clusters A and B (FDR < 0.0001). Furthermore, 24 genes with an FDR of <0.000001 and a log fold change (FC) of >3.9 were noted. Among microRNAs, *miR125B1* had the smallest FDR, followed by *miR1245A* (Appendix A). The expression of *miR125B1* and *miR1245A* was significantly upregulated in Cluster B (*p* < 0.0000001 by Mann–Whitney U test) (Figure 3a); the expression of *miR125B1* and *miR1245A* largely differed between the two clusters (log FC = 4.9 and 3.9, respectively).

To examine the biological significance of these miRNAs in GBC, we performed invasion and growth assays following knockout of these miRNAs in two GBC cell lines, NOZ and G415. These miRNAs in GBC cells were edited by CRISPR-Cas9 technology, and the edited cells were enriched by fluorescence-activated cell sorting (FACS) (Appendix A). Editing of the target miRNAs was confirmed by polymerase chain reaction (PCR) and amplicon sequencing, which showed *miR125B1* gene had deletions or insertions in edited NOZ cells and G415 cells (Figure 5a), but *miR1245A* had no change. We observed that the invasion ability of NOZ cells was significantly reduced following *miR125B1* knockout (*p* = 0.00124 by *t*-test) (Figure 5b left and Appendix A). G415 cells also showed a trend of reduction in their invasion ability (*p* = 0.14466, by *t*-test) (Figure 5b right). These results suggest that the inhibition of the non-coding gene *miR125B1* may result in the attenuation of the invasive potential of GBC cells and that *miR125B1* could involve the EMT of GBCs.

We further performed RNA-seq of NOZ and G415 cells with *miR125B1* knockout and analyzed the changes in the expression of RNAs. The single-sample GSEA (ssGSEA) analysis demonstrated the reduced activity of the genes involved in EMT and inflammatory response (Appendix A), consistent with the reduced invasion ability of GBC cells following *miR125B* knockout.

### 2.6. Mutational Profiles of GBC and Mutations of TGF-β Signaling Pathway Genes

*TP53* gene alterations are frequently noted in more than 30% of GBCs [8]. Mutations of *EGFR*, *ERBB3*, *PTEN*, and *ARID2* and the APOBEC mutational signature [21] were found to be specific in GBCs among diverse BTCs. In our exome/whole-genome sequencing (WGS) analysis of 39 GBCs (in addition to three GBCs), mutations of *TP53* (*n* = 18, 47%), *TTN* (*n* = 7, 18%), *ABCA13* (*n* = 5, 13%), *ELF3* (*n* = 5, 13%), and *ARID1A* (*n* = 4, 11%) were frequently noted (Figure 6a). Considering the differences between Clusters A and B, *ARID1A* was observed in four cases (11%) and *DST* and *ERBB3* in three cases (8%) specifically in Cluster B; however, the difference was not statistically significant. Based on the oncogenic signaling pathways revealed from The Cancer Genome Atlas (TCGA), we found that the mutations of the genes related to the TGF-β signaling pathway were enriched in Cluster B (Figure 6b). In total, 71% TGF-β signaling genes in Cluster B (5 of 7 related genes; *ACVR1B*, *ACVR2A*, *SMAD2*, *SMAD3*, *SMAD4*, *TGFBR1*, *TGFBR2*) and 28% in Cluster A (2 of 7) were mutated. *ELF3* is also reported to be associated with the TGF-β signaling pathway and EMT in some cancers [22]. On the other hand, the number of affected genes of the RTK–RAS and WNT signaling pathways were 20% (17 of 85 related genes) and 8.9% (6 of 68 related genes), respectively. These genomic and transcriptomic analyses suggest that the TGF-β signaling pathway and EMT are likely to drive GBC development and aggressiveness in Cluster B.

We analyzed the mutation profiles of 13 WGS and 26 exome sequences of GBCs and detected 28 types of COSMIC mutational signatures [23]. Of these, 23 GBCs showed COSMIC Sig2 and Sig13 abundance recognized as APOBEC signatures [24,25], and four samples had high amounts of APOBEC signatures (over 0.3) (Appendix A). Only one GBC (HK94) had a particularly large number of somatic mutations (1751 single nucleotide variations (SNVs)) and showed a high abundance of APOBEC signatures (COSMIC Sig2 was 0.678 and Sig13 was 0.313) (Appendix A). However, no somatic variation or transcriptomic change in *APOBEC* genes (*APOBEC3A*, *APOBEC3B*, *APOBEC3C*, *APOBEC3D*, *APOBEC3F*, *APOBEC3G*, and *APOBEC3H*) [25] was observed in this sample, indicating that the unknown factor involves the mutagenesis of APOBEC signatures in GBCs. These findings indicate that APOBEC signatures and the underlying mutagenesis may play some important roles in GBC development.

### 2.7. Copy Number Variants (CNVs) and Fusion Events in GBCs

We analyzed CNVs from 11 WGS and 26 whole-exome sequences’ data using Sequenza. GISTIC2 analysis revealed the gain in the copy numbers of 1p36.33, 7q22.1, 17q12, and 19q12 regions in GBCs (Appendix A). These copy number gains were not significantly related to patient prognosis, although a trend (*p* = 0.1) of short overall survival was noted with 19q12 gain that includes the *CCNE1* gene [26,27] (Appendix A).

Fusion genes in cancer are generated through somatic structure variants (SVs) and may serve as important therapeutic targets [28]. *FGFR 2/3* gene fusion has been reported in several tumor types, including intrahepatic cholangiocarcinoma, and these fusion events confer sensitivity to fibroblast growth factor receptor (FGFR) inhibitors [29]. RNA-seq analysis revealed many types of fusion events in GBCs. A GBC sample was found to exhibit *TARDBP–FGFR3* gene fusion, which was supported by more than 3000 reads (Appendix A). The expression level of *FGFR3* transcript in this tumor was much higher than that detected in other 35 cases (about 200 times in FPKM), indicating that it may constitutively activate *FGFR3* by changing its regulatory domain. *C15orf57–CBX3* fusion (11 cases; 30.6%) induced by SV at Chr15–Chr7 and *GAPDH–GPBP1* fusion (3 cases; 8.3%) induced by SV at Chr12–Chr5 were recurrently detected in GBCs. *C15orf57–CBX3* fusion was observed in 22.7% of HCCs, glioblastomas, melanomas, and lymphomas [30], although its functional significance is yet unknown.

## 3. Discussion

In the present study, we analyzed the comprehensive transcriptome and genome of GBCs in addition to HBDCs, which are anatomically close to and sometimes difficult to discriminate from CDCs or GBCs [13]. As a result, no large differences were observed between the transcriptomic profiles of GBCs and HBDCs (Appendix A). On the other hand, the transcriptome analysis classified GBCs into two subgroups based on the expression of the genes related to TME (stroma and inflammation). Cluster B, with higher abundance and activity of TME, showed poorer prognosis than Cluster A. One of the typical features of BTC and GBC is the highly desmoplastic microenvironment rich in the fibrogenic connective tissue [31]; thus, TME evolves together with the tumor mass, thereby restricting the delivery of several drugs [32]. Stromal fibroblasts from TME have been known to play tumor-supportive and -suppressive roles by dysregulating the wound-healing response [33]. In pancreatic cancer, some substances secreted by stromal cells stimulate cancer cell migration and invasion, inhibit apoptosis, and increase resistance to chemotherapy and radiation [34]. Advanced GBCs in Cluster B were similar to pancreatic cancer in terms of TME-rich tumor, and thus may be resistant to chemotherapies. Stromal components in TME were also considered to prevent the immune cell-mediated attack on tumor cells [35]. Some clinical trials have been conducted on emerging immuno-checkpoint inhibitors for BTC and GBC treatment, and their efficacy or benefits are expected in patients with advance tumors with immunologically “hot” and high tumor mutation burden [36]. However, our study shows that many immunosuppressive mechanisms, such as Tregs, M2 macrophages, *IL-10*, and *TGFB1*, were activated in GBCs of Cluster B, and whether only programmed cell death-1 (PD-1) blockade was effective or combination immunotherapies with TME modification are essential is yet questionable [36,37].

In this study, we analyzed the expression of ncRNAs that can affect the TME in GBC in addition to coding genes related to EMT or TGF-β. For instance, *miR-100* and *miR125B* expression was associated with TGF-β in pancreatic ductal adenocarcinoma and found to be upregulated in patients with poor prognosis [38]. Invasion assay results using GBC cells suggest that the inhibition of *miR125B1* may reduce the invasion ability of cells. As *miR125B1* upregulates EMT via TGF-β signaling [38], this result suggests that the inhibition of *miR125B1* may reduce EMT and cell invasion. The increase of stroma and vimentin-positive stromal cells of Cluster B GBCs indicates that fibroblasts in the stroma are induced to CAFs by various crosstalk between cancer cells and stromal cells in TME, including inflammation and the TGF-β and EMT pathways; then, this crosstalk in TME induces cancer aggressiveness and worse prognosis [39]. So, decreased invasion ability in cancer cells with *miR125B1* knockout means biologically decreased activity of the EMT pathway in cancer cells. Hence, *miR125B1* may be a new therapeutic target of BTC or a poor prognostic factor, because its expression was remarkably upregulated in Cluster B. The expression of *miR1245A* was similarly upregulated in Cluster B but had no effect on the invasion ability of cells, which was confirmed by the experiments with edited cells. This result is consistent with the fact that *miR1245A* is associated with DNA repair and cell apoptosis [40] and had no effect on cell invasion and EMT. However, it may serve as a biomarker of poor prognosis in GBC.

Genomic analysis revealed more mutations of the TGF-β signaling pathway in Cluster B than in Cluster A, in addition to the upregulation in the expression of the TGF-β signaling pathway genes in Cluster B, as evident from GSEA. Alteration of the TGF-β signaling pathway induces EMT and increases tumor aggressiveness and is central to immune suppression within TME [14,41]. GBCs in Cluster B expressed many types of immune checkpoints or suppressor molecules, such as PD-1, CTLA4, and TIM-3. Further, the function of CD8^+^ effector T cells was suppressed, although these cells were abundantly present in the stroma. The TGF-β signaling pathway plays a central role in the poor prognosis and high aggressiveness of GBCs, and it is possible that therapeutics targeting the TGF-β signaling pathway would be effective for GBC treatment [42].

In summary, this gene clustering analysis of 36 GBCs reveals there is a subgroup with a poor prognosis even at early stages, and it is characterized as TME-rich GBCs, which have the increased expression of EMT-related genes and TGF-β pathway alterations. This demonstrated that TME, EMT, and TGF-β pathway alterations are the main drivers of GBC. The limitation of this study is that the sample size of GBCs was still not large enough, and considering the racial difference of GBC etiology [1,42], we should analyze more GBC samples from diverse populations, including European and African populations, where GBC is very rare.

## 4. Materials and Methods

### 4.1. Patients and Samples

We obtained 37 fresh-frozen and two PAXgene^®^-embedded tumor specimens from patients with GBCs and BTCs who underwent surgery at the Hokkaido University Hospital between 2004 and 2018. Clinical features of 39 patients with GBC and 8 HBDCs are shown in Appendix A. DNA of 38 GBCs samples was sequenced in our previous paper [21]; however, we called their mutations by different methods. RNA and DNA were extracted from all cells taken from the surface to the invasive area in tumor blocks. All patients provided written informed consent, and the study was approved by the institutional review boards of Hokkaido University and RIKEN.

### 4.2. Library Preparation and DNA or RNA Sequencing

Exome capture technique was performed using Nextera Rapid Capture Exomes kits (Illumina, San Diego, CA, USA). For WGS, the DNA was extracted from cancer and normal tissues and 500–600-bp insert libraries were prepared according to the protocol provided by Illumina. The exome-captured or WGS libraries were sequenced on HiSeq2500 with paired reads of 125 bp or NovoSeq6000 with paired reads of 150 bp. Total RNA was extracted from 37 fresh-frozen tumor specimens; one sample was excluded from RNA sequencing, owing to its very low RNA integrity number (RIN) score. We used the KAPA RNA HyperPrep Kit with RiboErase (Illumina, San Diego, CA, USA) for RNA-seq library preparation. Poly(A) selection, cDNA synthesis, and library construction were performed according to the manufacturer’s protocols. Sequencing was carried out on the HiSeq2500 platform, and STAR aligner (ver. 2.5.3a, https://github.com/alexdobin/STAR, accessed on 1 December 2020) was used to map RNA-seq reads onto the reference human genome GRCh37. The featureCounts (https://rdrr.io/bioc/Rsubread/man/featureCounts.html, accessed on 1 December 2020) software was employed to count reads per gene using GENCODE release 19 as gene model. FPKM was computed using an in-house R script.

### 4.3. Statistical Analysis

Kaplan–Maier survival analysis, Cox proportional hazard regression analysis, and Mann–Whitney U-test were performed via the R packages Survival and Coin.

### 4.4. GSEA

The h.all.v6.2.sytmbols.gmt in the Molecular Signatures Database (MSigDB) was selected as the reference gene set, and an FDR-adjusted value of <0.25 was chosen as the cut-off criterion. ssGSEA for knockout (KO) cell lines was performed using GenePattern ssGSEA projection (https://www.genepattern.org/, accessed on 1 December 2020) as the reference gene set of MSigDB (https://data.broadinstitute.org/gsea-msigdb/msigdb/release/7.1, accessed on 1 December 2020). 

### 4.5. SNV, INDELs, and Fusion Calling

For the analysis of SNV and INDELs, the Genomon2 (ver2.6.1, https://genomon.readthedocs.io/ja/v2.6.1/, accessed on 1 December 2020). Fisher mutation call was employed. The minimum depth for the mutation call was 8, while the minimum map quality was 20; the minimum base quality was 15. Variants with less than 3 reads were filtered. The disease minimum allele frequency chosen for our study was 0.05, while the control maximum allele frequency was 0.1. The Fisher mutation call threshold was set to 0.05. The detected variants were annotated using ANNOVAR [43] and summarized and visualized using the R package Maftools (https://doi.org/doi:10.18129/B9.bioc.maftools, accessed on 1 December 2020) [44].

Exome sequencing and WGS data were processed as follows: After alignment of reads using BWA [45], PCR duplicates were removed using Picard (http://picard.sourceforge.net/, accessed on 1 December 2020), and SNVs and Indels were detected using Genomon2. RNA sequence fusion was detected using Genomon2.

### 4.6. DNA Copy Number Analysis

For analysis of CNVs from exome sequencing and WGS, we used the R package Sequenza (https://cran.r-project.org/web/packages/sequenza/vignettes/sequenza.html, accessed on 1 December 2020). Copy number alterations were analyzed using GISTIC2 (ver7, https://www.genepattern.org/modules/docs/GISTIC_2.0, accessed on 1 December 2020).

### 4.7. Immunohistochemistry

Tissue sections were deparaffinized in xylene and rehydrated in graded ethanol series. Heat-induced antigen retrieval was performed in a high- or low-pH antigen retrieval buffer (DakoCytomation, Glostrup, Denmark). Endogenous peroxidase activity was blocked by incubation of tissue sections in 3% hydrogen peroxide for 5 min. Primary antibodies against CD8 (1:500, IR6231-2; Dako), vimentin (1:500, IR63061-2J; Dako), and TIM-3 (1/400, AF2365, R&D Systems) were applied for 30 min. The sections were visualized using the horseradish peroxidase (HRP)-labeled polymer method (EnVision FLEX System, Dako). Immunostained sections were counterstained with hematoxylin, dehydrated in ethanol, and cleared in xylene.

### 4.8. Cell Culture

The NOZ (PRID: CVCL_3079) cell line was purchased from the Japanese Collection of Research Bioresources (JCRB) Cell Bank (Tokyo, Japan) in 2019, and the G415 (PRID: CVCL_8198) cell line was obtained from the RIKEN BRC Cell Bank (Japan)/Tohoku University in 2019. NOZ cells were maintained in William’s medium E supplemented with 10% fetal bovine serum, penicillin (100 U/mL), and streptomycin (100 µg/mL; Thermo Fisher Scientific, Waltham, MA, USA). G-415 cells were cultured in Roswell Park Memorial Institute (RPMI)-1640 medium supplemented with 10% fetal bovine serum, penicillin (100 U/mL), and streptomycin (100 µg/mL). Both cell lines were incubated in the presence of 5% circulating CO_2_ at 37 °C. All experiments were performed with mycoplasma-free cells.

### 4.9. Generation of CRISPR-Cas9-Mediated KO Cell Lines

The single-guide RNAs (sgRNAs) for CRISPR-mediated KO cell line generation were designed using the CRISPR design tool (https://crispr.dbcls.jp/, accessed on 1 December 2020). Oligonucleotides were synthesized by Integrated DNA Technologies (IDT) and their sequences are provided in Appendix A. We introduced the CRISPR protein (Alt-R^®^ S.p. Cas9 Nuclease V3, IDT, Redwood City, CA, USA) and sgRNAs by electroporation method using the protocol provided by IDT. The setting for electroporation was provided by LONZA (DN100). We also simultaneously introduced ATTO550 transactivating RNA (trRNA), an RNA molecule labeled with a fluorescent dye, into cells with CRISPR and enriched the fluorescently labeled cells using FACS. After enrichment, cells were incubated at 37 °C in a 5% CO_2_ incubator for 48 h. DNAs and RNAs were extracted from transfected cells and evaluated for genomic locus edit by CRISPR-Cas9 using PCR primers. The sequence was indicated in Appendix A. The mutations edited by CRISPR were confirmed by amplicon sequencing in MiSeq sequencer (Illumina, San Diego, CA, USA).

### 4.10. Invasion and Cell Viability Assays

The invasion assay was performed using Corning^®^ BioCoat™ Matrigel^®^ Invasion Chamber (CORNING, Glendale, AZ, USA. 24-well plates, 8.0 microns). Cell lines without prestarvation treatment were cultured for 24 h at 37 °C in a 5% CO_2_ incubator, fixed, and stained using Diff-Quik (Sysmex, Kobe, Japan) following the manufacturer’s protocols. Cell viability was evaluated using the Cell Counting Kit-8 (DOJINDO, Kumamoto, Japan) as per the manufacturer’s protocol. After incubation, absorbance was measured at 450 nm/620 nm using a microplate reader. The triplicate of the average of the calculated difference between the test measurement and reference measurement was used as the OD. We calculated cell/OD as an index of invasion per unit area corrected by the number of seeded cells. The membrane was imaged by dividing it into four pieces under a 4× objective lens, and the number of cells was counted.

### 4.11. RNA-Seq Analysis of the KO Cells

The RNA was extracted from 70–80% confluent cells by RNeasy Plus Mini Kit^®^ (QIAGEN, Venlo, The Netherlands), and the KAPA RNA HyperPrep Kit with RiboErase (Illumina, San Diego, CA, USA) was used for RNA-seq library preparation. Library sequencing and analysis were performed in the same manner as RNA extraction from tissues.

### 4.12. Data Deposition

WGS and exome sequencing data in this study were deposited in NBDC (National Bioscience Database Center) under the accession numbers JGAD00000000117, JGAD00000000118, hum0103, and hum0158. RNA-seq data in this study were deposited in DDBJ (DNA Data Bank of Japan) Center under the accession numbers SAMD00254734–SAMD00254781.

## 5. Conclusions

This comprehensive analysis demonstrated that TME, EMT, and TGF-β pathway alterations are the main drivers of GBC and provides a new classification of GBCs based on the status of TME, which is involved with EMT and immune suppression for poor prognosis of GBC. This information may be useful for GBC prognosis and therapeutic decision-making.

## Figures and Tables

**Figure 1 cancers-13-00733-f001:**
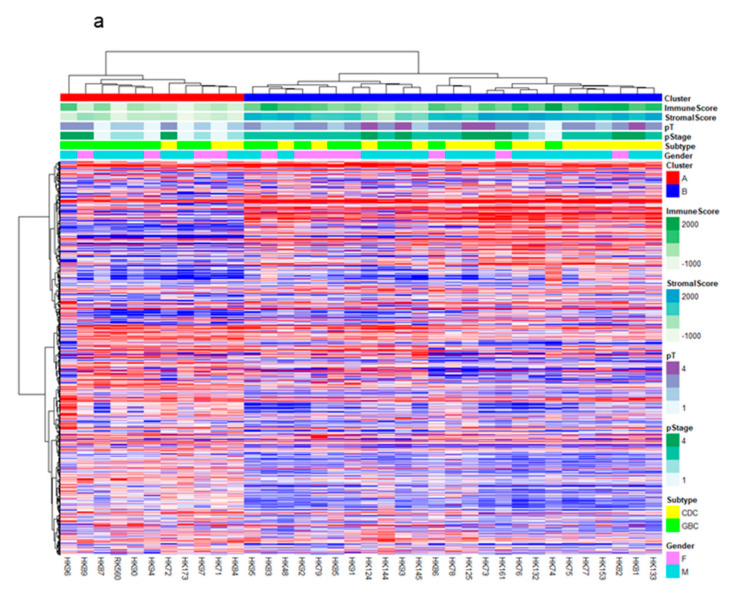
Molecular subclass of GBCs and their prognosis. (**a**) Hierarchical clustering analysis of the protein-coding gene expression data from 36 gallbladder cancer (GBC) samples; heatmap clustering of differentially expressed protein-coding genes. These were classified into two groups, Clusters A and B. Stromal score and immune score calculated by “Estimate” correlated with these clusters. (**b**) Kaplan–Meier plots of disease-free survival (left) and overall survival (right) of resected GBC cases. Significant differences in the prognosis between Clusters A and B are observed (*p* = 0.01 and 0.005 by log-rank test, respectively, and HR: 6.6439, 95% CI (1.514–29.16) and HR: 3.623, 95% CI (1.226–10.71)). Each table in the bottom shows numbers at risk.

**Figure 2 cancers-13-00733-f002:**
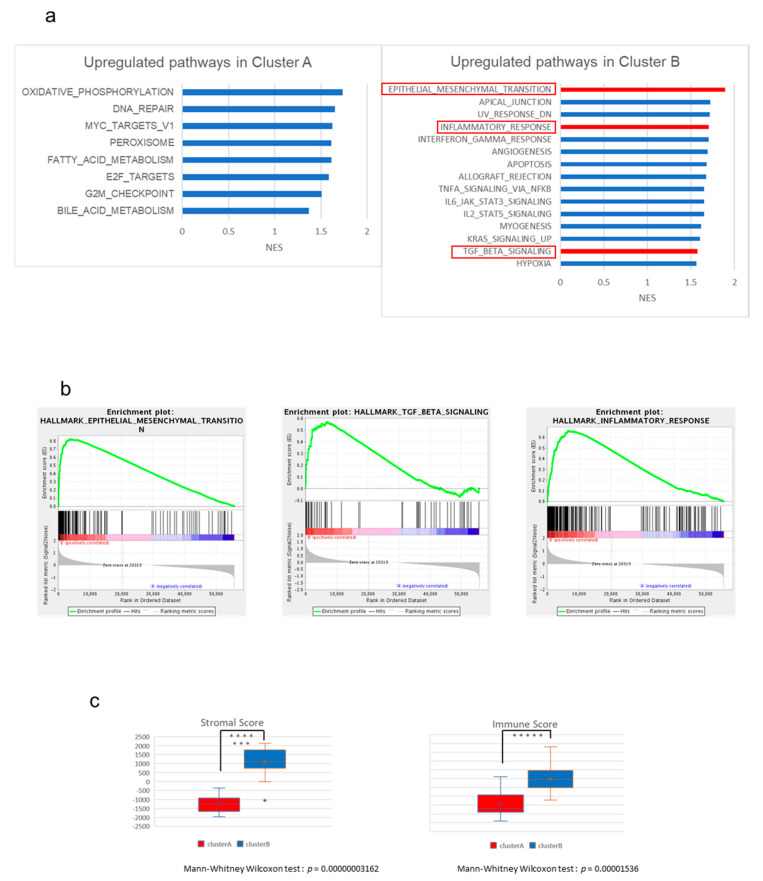
The results of gene set enrichment analysis (GSEA) between Clusters A and B. (**a**) All upregulated gene sets in Cluster A (left) and top 15 upregulated gene sets in Cluster B (right). (NES > 1.37 and > 1.18). (**b**) Gene sets of “EMT”, “TGF-β pathway”, and “inflammatory response” were significantly upregulated in Cluster B. (**c**) Comparison of the two groups based on stromal score (left) and immune score (right) calculated by “Estimate” (Mann–Whitney Wilcoxon test, *p* < 0.05). The *y*-axis shows each “Estimate” score. When *p* < 0.05, the asterisk increases with each zero digit; for example, *** for *p* < 0.0001.

**Figure 3 cancers-13-00733-f003:**
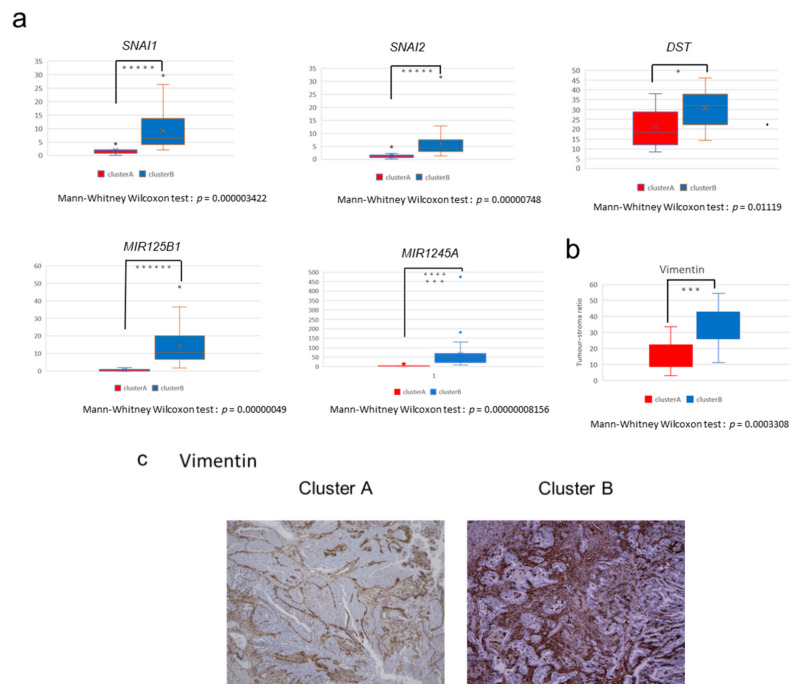
Expression of stroma-related genes in GBCs. (**a**) Comparison of the expression of stroma-related genes (*SNAI1*, *SNAI2*, *DST*, *miRNA125b1*, and *miRNA1245A*) between the two clusters in fragments per kilobase of exon per million fragments mapped (FPKM; Mann–Whitney Wilcoxon test, *p* < 0.05). The *y*-axis shows FPKM of each gene. When *p* < 0.05, the asterisk increases with each zero digit; for example, *** for *p* < 0.0001. (**b**) Vimentin immunostaining. Ratio of the area of the positive cells/tumor cells was used to calculate stromal ratio in tumors (*p* = 0.0003308 by Mann–Whitney U test). (**c**) Vimentin immunostaining in the representative GBC tissues in Cluster A (left) and Cluster B (right). Positive cells are brown in color. Each sample was observed with a 4× objective lens.

**Figure 4 cancers-13-00733-f004:**
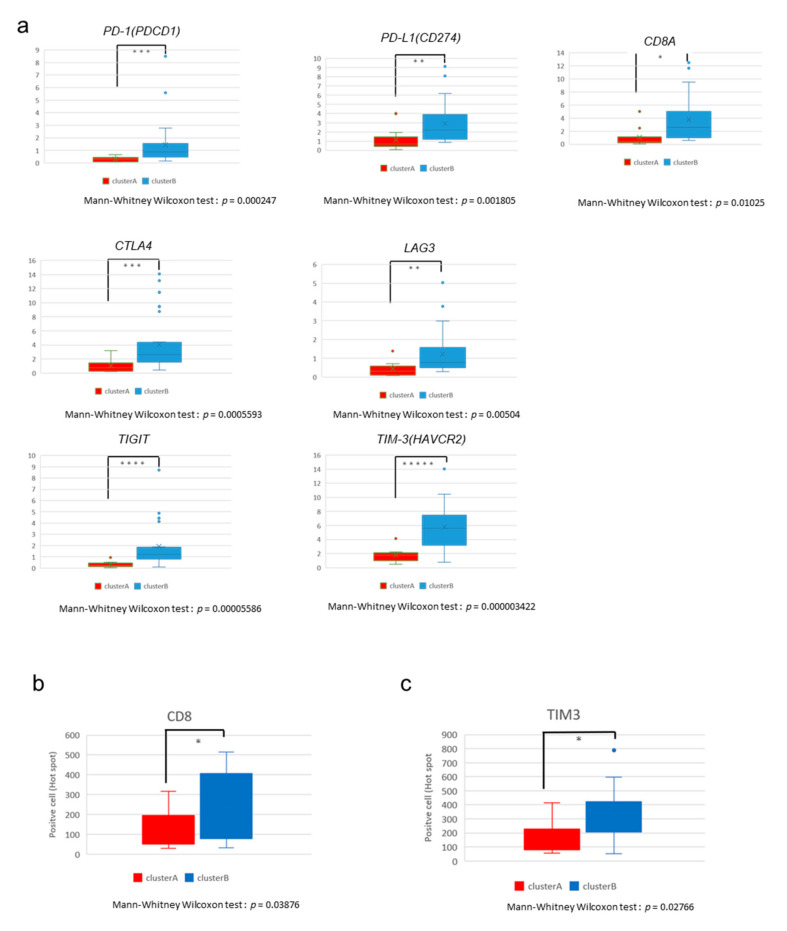
Expression of immune genes in GBCs. (**a**) Comparison of immune genes (*PD-1*, *PD-L1*, *CD8A*, *CTLA4*, *LAG3*, *TIGIT*, and *TIM-3*) between the two clusters in FPKM (* Mann–Whitney Wilcoxon test, *p* < 0.05). The *y*-axis shows the FPKM of each gene. When *p* < 0.05, the asterisk increases with each zero digit; for example, *** for *p* < 0.0001. (**b**) The number of CD8-positive cells at “hot spot” areas within the same magnification (*p* = 0.03876 by Mann–Whitney U test). (**c**) The number of TIM-3-positive cells at “hot spot” areas within the same magnification (*p* = 0.02766 by Mann–Whitney U test). (**d**) CD8 immunostaining in the “hot spots” of the representative GBC tissues in Cluster A (left) and Cluster B (right). Positive cells are brown in color. Each sample were observed with a 10× objective lens. (**e**) TIM-3 immunostaining in the “hot spots” of representative GBC tissues in Cluster A (left) and Cluster B (right). Each sample was observed with a 10× objective lens.

**Figure 5 cancers-13-00733-f005:**
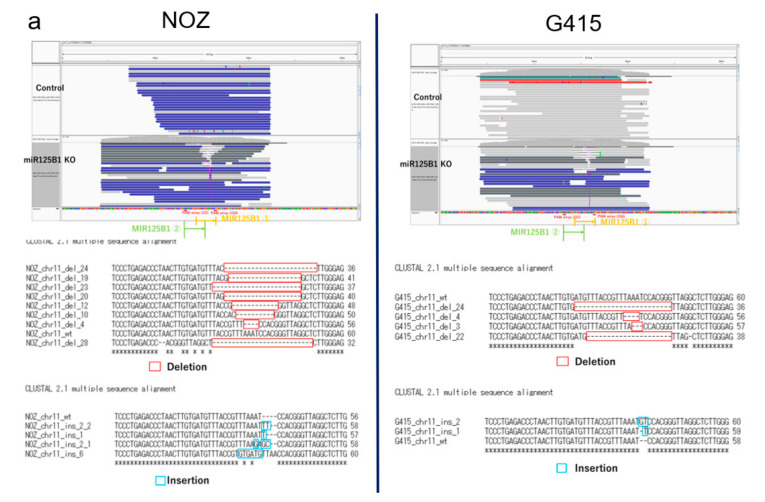
Knockout of *miR125B* in GBC cell lines. (**a**) The genomic region of *miR125B* was edited by CRISPR technology. This region was amplified by PCR from DNA of the edited and control cells of NOZ (left) and G415 (right) and sequenced by MiSeq. The IGV pictures are shown (upper). Deletion or insertion reads in *miR125B* were observed (bottom). (**b**) Cells that invaded through the Matrigel were counted three times to calculate average cells per membrane. The triplicate of the average of the calculated difference between the test measurement and reference measurement was used as the OD. We calculated cell/OD as an index of invasion per unit area corrected by the number of seeded cells, to compare cell/OD values between *miR125B1* knockout NOZ and G415 cell lines (*p* = 0.00124 and 0.14466 by *t*-test, respectively).

**Figure 6 cancers-13-00733-f006:**
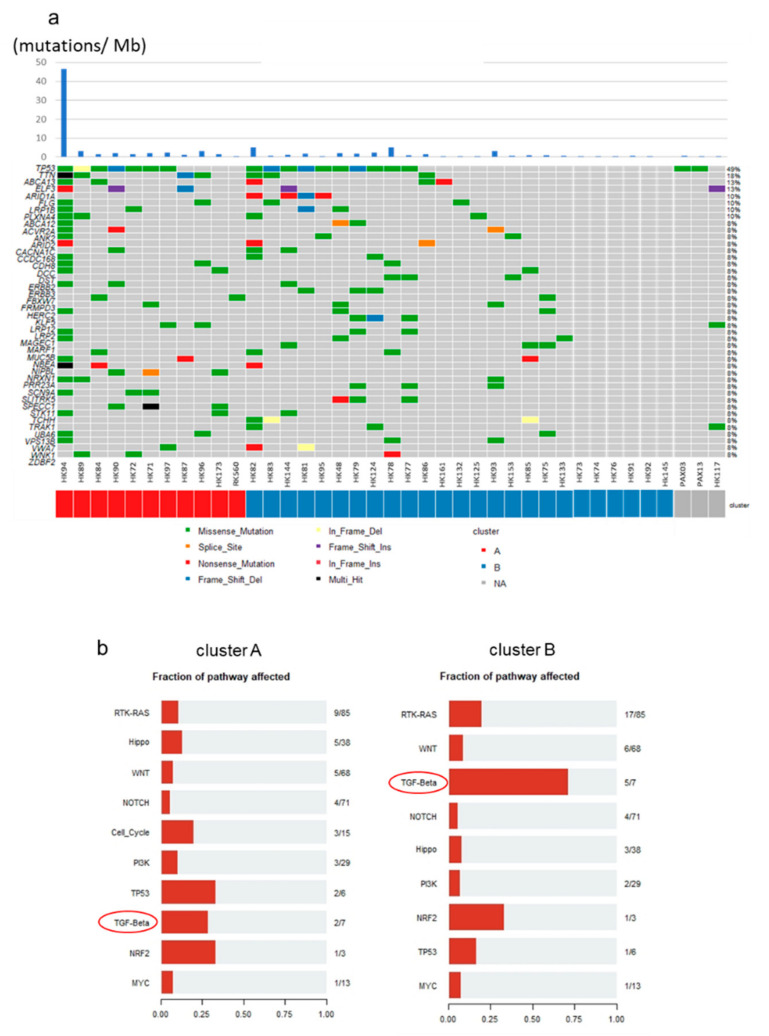
Mutational profiles of GBCs. (**a**) Landscape of somatic mutations of 39 GBCs in Clusters A and B. Three samples had no RNA-seq data and were not classified. Single nucleotide variations (SNVs)/INDELs across 39 GBC patients with different mutation types are coded by different colors. The upper panel shows the somatic mutation number per Mb. (**b**) Fraction of pathways affected; this figure was prepared from the number of related genes in each cluster. Each pathway-related gene set was used, as revealed from The Cancer Genome Atlas (TCGA). In total, 5 of 7 (71.4%) and 2 of 7 (28.6%) genes related to the TGF-β pathway from Cluster B and A had mutations, respectively.

## Data Availability

WGS and exome sequencing data in this study were deposited in NBDC (National Bioscience Database Center) under the accession numbers JGAD00000000117, JGAD00000000118, hum0103, and hum0158. RNA-seq data in this study were deposited in DDBJ (DNA Data Bank of Japan) Center under the accession numbers SAMD00254734–SAMD00254781.

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
