# Peer review of "Molecular Classification and Tumor Microenvironment Characterization of Gallbladder Cancer by Comprehensive Genomic and Transcriptomic Analysis"

_cancers, 2021, doi:10.3390/cancers13040733_

Round 1

Reviewer 1 Report

Dear authors,

thank you for your thorough review. The work has improved and can be accepted, in my opinion, for publication in this new revised form

Author Response

>Thank you for your positive comments.

Reviewer 2 Report

The manuscript is revised according to this reviewer’s comments, however, there still remain several concerns, as follows;

1) Carcinoma cells (or epithelial cells?) did not show positive immunoreactivity for vimentin, which show EMT. Since GBC cells did not show EMT, it may be inconsistent that the biological significance of miR125B1 and MIR1245A in terms of EMT was examined in GBC cells. Does the description “fibroblast cells are induced by activated EMT pathway of cancer cells” in authors’ replay mean cancer cells changed into fibroblasts via EMT pathway? If so, it is generally incredible and authors may show evidences to support authors’ hypothesis.

Author Response

>Thank you for your suggestion. We have some confusions about vimentin. We analyzed vimentin expression as a marker of cancer stromal fibroblasts (CAFs) in the stroma, which indicates stroma-richness and TME activity.

We changed the sentence below to explain about vimentin in Results2.4;

“Vimentin is a marker of mesenchymal-derived cells and CAFs (cancer-associated fibroblasts) which involve cancer stroma component and TME [17].”

We changed the description the reviewer suggested to the sentence below in Discussion;

“The increase of stroma and Vimentin-positive stromal cells of Cluster B GBCs indicates that fibroblast in the stroma is induced to CAFs by various crosstalk between cancer cells and stromal cells in TME, including inflammation, TGF-β and EMT pathways, then these crosstalk in TME induces cancer aggressiveness and worse prognosis [39].”

Following these changes, we also changed the references 17 and 41.

Reviewer 3 Report

The authors addressed all my relevant review points. There is just one of my minor critics not sufficiently answered ("Minor points: Please add numbers at risk to the survival analysis"). This refers to figure 1b. 

I recommend publication.

Author Response

>We misunderstood “number at risk”.  We added Tables of number at risk in Figure 1b with KM curves.

Round 2

Reviewer 2 Report

The manuscript is well revised according to this reviewer's suggestion.

This manuscript is a resubmission of an earlier submission. The following is a list of the peer review reports and author responses from that submission.

Round 1

Reviewer 1 Report

In this work, the authors performed a genomic and transcriptomic analyses on 36 GBC samples derived from Japanese patients. Through a clustering analysis they identify two subgroups of GBCs that are different for the expression of genes related to the tumor microenvironment. The study is important despite the limited case series and emphasizes the need to better characterize tumors of the biliary tract, not only by anatomical site, but also on the basis of their molecular and genetic profile, as already documented for intrahepatic cholangiocarcinomas.

However the work needs some revisions

1) Tumors of the biliary tract, including those of the gallbladder, have a different etiology in different countries, depending on ethnicity, and also have different molecular characteristics. The case series is limited, but it would be interesting to verify if the profiles identified in this study on a Japanese patients reflect those (on public databases) of other case series, of other ethnic groups, Western for example.

2) In line 103 Fig. A1 is indicated, but in supplementary material it is indicated as Figure 1. Please correct.

3) In supplementary material Table A1 it is also indicated with two titles (page S8). Please, delete the first. Furthermore, in the same table A1, GBBC and HBDC are indicated in the abbreviations, but these are not included in the table. Only CDC and GBC appear.

4) Two titles also appear in Table A2

5) In table A3 there is a double title, but they are different for FDR. The text indicates FDR < 0.25. Which of the two values is correct?

6) In table A4, previously indicated as table 4, there are always two titles. In table A5 and A6, the same problem. I suggest for all the tables, in the supplementary material, to leave only one title in each case, the more explanatory one, at the top.

7) On page S12 there is a table indicating the sequence of primers. Is it table A7? Please indicate it and add the title.

8) Figure 2: I would change the position of the two panels. Cluster A on the left and cluster B on the right

9) Figure 4 does not indicate the letter a in the panel (page 7)

10) In many figures (Figures 2c, 3a, 4a, 5b) in the y axis it is not indicated what it represents.

11) Even in the supplementary material (Figure A2, A5) in the y axis it is not indicated what it represents.

12) Figure 3c is not mentioned in the text.

13) As regards the immunohistochemical images (Figure 3c, 4d and 4e), I suggest replacing with representative images, not with different magnification of the same positivity, but three different samples with three different intensities for vimentin, CD8 and TIM-3 respectively.

14) In figures A2 in Supplementary material (S2) add, in the legend, the full abbreviations of CDC and GBBD.

15) miR125B1 and miR1245A, both are statistically significantly deregulated. Justify why it was chosen to study only the role of miR125B1.

16) In figure 5, I would delete the panel a, and I would replace it with a panel related to the expression level of miR15B1 after knock-out in the two cell lines.

17) Please, indicate in the legend how many invasion and viability assays experiments were carried out.

Author Response

Reviewer 1

In this work, the authors performed a genomic and transcriptomic analyses on 36 GBC samples derived from Japanese patients. Through a clustering analysis they identify two subgroups of GBCs that are different for the expression of genes related to the tumor microenvironment. The study is important despite the limited case series and emphasizes the need to better characterize tumors of the biliary tract, not only by anatomical site, but also on the basis of their molecular and genetic profile, as already documented for intrahepatic cholangiocarcinomas.

However the work needs some revisions

1)Tumors of the biliary tract, including those of the gallbladder, have a different etiology in different countries, depending on ethnicity, and also have different molecular characteristics. The case series is limited, but it would be interesting to verify if the profiles identified in this study on a Japanese patients reflect those (on public databases) of other case series, of other ethnic groups, Western for example.

>> Thank you for your suggestion. We agree that GBCs have a different etiology in different countries, depending on ethnicity, and the etiological difference in the molecular level is also interesting. But usually mutational signature analysis and DNA methylation analysis of DNA level are the best approaches to understand the etiological difference of cancer genome, and we think that RNA expression analysis can be the second approach. 

GBC is a rare tumor in Europe/USA and there are only one case of GBC deposited in the most common public databases, TCGA and ICGC. Recently, one study analyzed 100~ GBC cases (Pandey A et al. Nat Commun. 2020, 11, 4225), but most of these GBC cases are Indian or Korean, not European, and these data is not available in public. We cited this paper as a reference in Introduction and add a comment about etiological diversity of GBC in Discussion as a limitation of this study.

“However, the limitation of this study is that the sample size of GBC is still not enough and, considering the racial difference of GBC etiology [1][41], we should analyze the comprehensive DNA/RNA data of more GBC samples from diverse populations including European and African populations where GBC is very rare.”

2) In line 103 Fig. A1 is indicated, but in supplementary material it is indicated as Figure 1. Please correct.

>> Figure A1 shows the clustering analysis of GBC and HBDC and we described this at the first paragraph of 2.1 session.

“The clustering analysis of RNA-seq data revealed no apparent difference between HBDCs, CDCs, and GBBCs (Fig. A1), indicating their similar molecular features.”

3) In supplementary material Table A1 it is also indicated with two titles (page S8). Please, delete the first. Furthermore, in the same table A1, GBBC and HBDC are indicated in the abbreviations, but these are not included in the table. Only CDC and GBC appear. 

>> We added clinical information of 8 HBDC cases in Table A1.

4) Two titles also appear in Table A2 

>>Thank you for your suggestion. We corrected this.

5) In table A3 there is a double title, but they are different for FDR. The text indicates FDR < 0.25. Which of the two values is correct? 

>> Table A3 shows the list of pathways with FDR<0.25. We corrected this issue in the text and Table A3.

6) In table A4, previously indicated as table 4, there are always two titles. In table A5 and A6, the same problem. I suggest for all the tables, in the supplementary material, to leave only one title in each case, the more explanatory one, at the top.

>> We also corrected the titles of all Tables.

7) On page S12 there is a table indicating the sequence of primers. Is it table A7? Please indicate it and add the title. 

>> Table 6A has the information of sgRNA (upper) and PCR primers (lower). We corrected them in the Supplementary data and Method in the text.

8) Figure 2: I would change the position of the two panels. Cluster A on the left and cluster B on the right

>> We put Cluster A on the left and cluster B on the right in Figure 2.

9) Figure 4 does not indicate the letter a in the panel (page 7) 

>> We added the letter (a) in Figure 4.

10) In many figures (Figures 2c, 3a, 4a, 5b) in the y axis it is not indicated what it represents.

>> In the figure legends, we added sentences to explain the y-axis and x-axis.

11) Even in the supplementary material (Figure A2, A5) in the y axis it is not indicated what it represents.

>> In the supplementary figure legends, we added sentences to explain the y-axis and x-axis.

12) Figure 3c is not mentioned in the text. 

>> We added Figure 3b and 3c in 2.4 of the text.

“Immunohistochemical staining for vimentin showed positive staining for stromal cells, not tumor cells, and significantly higher vimentin expression in GBCs of Cluster B than in GBCs of Cluster A (p = 0.0003308 by Mann–Whitney U test, Fig. 3b and 3c).”

13) As regards the immunohistochemical images (Figure 3c, 4d and 4e), I suggest replacing with representative images, not with different magnification of the same positivity, but three different samples with three different intensities for vimentin, CD8 and TIM-3 respectively.

>> We replaced them with new IHC image sets with the same magnitude of each cluster (A and B) for vimentin (Fig. 3C), CD8 (Fig 4d), and TIM3 (Fig 4e).

14) In figures A2 in Supplementary material (S2) add, in the legend, the full abbreviations of CDC and GBBD.

>> We added the full abbreviations of CDC and GBDC in the legend of Fig A2.

15) miR125B1 and miR1245A, both are statistically significantly deregulated. Justify why it was chosen to study only the role of miR125B1.

>> We performed invasion assays following knockout of miR1245A as well, but we did not confirm any changes of invasion ability. That is why we did not show the results of cell experiments. We commented this issue in Discussion.

“The expression of miR1245A was similarly upregulated in Cluster B but had no effect on the invasion ability of cells was confirmed by the experiments of edited cells. This result is consistent with the fact that miR1245A is associated with DNA repair and cell apoptosis [39] and had no effect on cell invasion and EMT. However, it may serve as a biomarker of poor prognosis in GBC.”

16) In figure 5, I would delete the panel a, and I would replace it with a panel related to the expression level of miR15B1 after knock-out in the two cell lines.

>> We moved this figure to supplementary figure (new Figure A6a) and moved the mutation status (Fig A6b) to new Figure 5a. We did not check expression of these miRNA, but we confirmed mutations of miRNA after CRISPR-editing. In the text in this session, we changed some words “miR125B1 expression” to “miR125B1 knockout”.

17) Please, indicate in the legend how many invasion and viability assays experiments were carried out.

>> We performed the invasion assays and viability assays three times and counted the cells three times for each experiment. We added this issue in the figure legend (Figure 5b) and Methods.

“Cells that invaded through the Matrigel were counted three times to calculate average cell per membrane. The triplicate of the average of the calculated difference between the test measurement and reference measurement was used as OD.”

Reviewer 2 Report

In this study, comprehensive genomic and transcriptomic analysis were performed in gallbladder cancer and high volume of data were presented. The weakness of this study is a limited number of patients (39 patients). Although two subclasses: Cluster A and Cluster B were proposed in this study, fair comparison should be performed between matched clinical stages. In fact, cluster A included much more patients with stages 1 and 2. Therefore, the difference between subclasses seems to depend on the stages, the extent of invasion, not on different characteristics of GBC carcinoma. There are several other points to be addressed as follows;   

Major

1) Size of tumor, histological subtypes, classification (wel, mod, por) should be included in supplementary table 1.

2) The sites from where the sample was obtained (intracystic/surface areas? invasive area?) should be described. The percentage of carcinoma cells in the sample should be also described.

3) Figures 4d and 4e: Figures of cluster A appear to be ICPN (WHO classification of digestive system tumours 2019). Histology of GBC may be re-evaluated in terms of association of ICPN, which may highlight some molecular differences in this comprehensive analysis.

4) Figure 3C: The figures of cluster A appear to be non-neoplastic liver with several portal tracts, not GBC tissue. The figures of cluster B appear to be non-invasive carcinoma or hyperplastic lesion.

5) Carcinoma cells (or epithelial cells?) did not show positive immunoreactivity for vimentin, which show EMT. Since GBC cells did not show EMT, it may be inconsistent that the biological significance of miR125B1 and MIR1245A in terms of EMT was examined in GBC cells.

Minor

1) If the data on mutation analysis was partly included in authors’ previous study (J Hepatol 2018), it should be noted in the manuscript.

2) Figure legends for Figure A4 and A5 are wrong.

3) Figure A6(c): Matrigel invasion assay; there seems to be few differences between the Non-editting cells and miR125B1 KO cells

Author Response

Reviewer 2

In this study, comprehensive genomic and transcriptomic analysis were performed in gallbladder cancer and high volume of data were presented. The weakness of this study is a limited number of patients (39 patients).

>> GBC is a rare type of cancer even in Japan. Usually it is diagnosed at a very advanced stage and resectable cases are also very rare. Hokkaido University Hospital is one of the highest volume centers for BTC in Japan, who resects about 3-4 GBC cases a year, and we collected 39 fresh-frozen surgical specimens for 15 years. Even TCGA had only one case of GBC, and this comprehensive data of 39 GBCs are valuable, we believe. We added the sentences below in Discussion as a limitation of this study.

“The limitation of this study is that the sample size of GBC is still not enough and, considering the racial difference of GBC etiology, we should analyze the comprehensive DNA/RNA data of more GBC samples from diverse populations including European and African populations where GBC is very rare.”

Although two subclasses: Cluster A and Cluster B were proposed in this study, fair comparison should be performed between matched clinical stages. In fact, cluster A included much more patients with stages 1 and 2. Therefore, the difference between subclasses seems to depend on the stages, the extent of invasion, not on different characteristics of GBC carcinoma.

>> In our multivariate analysis, OS depended on pT factor but DFS depend on pN factor and Cluster A/B, although sample numbers of each clinical groups are small, especially early stage ones. When calculating with Fisher’s exact test, we calculated them collectively divided into two clinically significant depths groups and thought that the cluster was a different factor from T. Table A2 shows the difference of gender, CDC/GBBC, pT (T1+T2 vs T3+T4), and pN+ between Clusters A and B did not reach statistical significance (p >0.05), suggesting significant enrichment of tumor stage in these clusters. We added the data of the multivariate analysis in Table A2 and describe this in the text and Methods.

“Cluster B showed significantly poorer prognosis and higher recurrence rate (p = 0.01 and 0.005 by log-rank test, respectively) with HR: 6.6439, 95%CI [1.514-29.16] and HR: 3.623, 95%CI [1.226-10.71] than Cluster A (Fig. 1b). However, the difference of gender, CDC/GBBC, pT (T1+T2 vs T3+T4), and pN+ between Clusters A and B did not reach statistical significance (p >0.05 by Fisher’s exact test, Table A2). The Cox proportional hazard regression analysis also showed that the overall survival (OS) depended on pT factor and the disease-free survival (DFS) depend on pN factor and Cluster A/B (Table A2), suggesting that Cluster A/B could be an independent factor associated with GBC prognosis.”

There are several other points to be addressed as follows;   

Major

1) Size of tumor, histological subtypes, classification (wel, mod, por) should be included in supplementary table 1.

>> We add histological subtypes and classification in Table A1. But size of tumor were not included, because tumor depth more contributes tumor staging in BTC than tumor size and the pT factor information represents this issue.

2) The sites from where the sample was obtained (intracystic/surface areas? invasive area?) should be described. The percentage of carcinoma cells in the sample should be also described.

>> RNA and DNA were extracted from all cells taken from the surface to the invasive area in tumor blocks. We add a comment about this issue in Methods.

Tumor purity evaluation of fresh-frozen samples is sometime difficult. FFPE block was created from the other part of tumor and tumor purity of FFPE does not always reflect tumor purity of fresh-frozen part. We calculated “Estimate” score from RNA-seq data of fresh-frozen samples, which provides with tumor purity information. We add tumor purity score calculated by Estimate in Table A1

3) Figures 4d and 4e: Figures of cluster A appear to be ICPN (WHO classification of digestive system tumours 2019). Histology of GBC may be re-evaluated in terms of association of ICPN, which may highlight some molecular differences in this comprehensive analysis.

 >> ï¼·ï½… confirmed this is not ICPN, and invasive cancer by our pathologists. We replaced them with new IHC image sets with the same magnitude of each cluster (A and B) for vimentin (Fig. 3C), CD8 (Fig 4d), and TIM3 (Fig 4e).

4) Figure 3C: The figures of cluster A appear to be non-neoplastic liver with several portal tracts, not GBC tissue. The figures of cluster B appear to be non-invasive carcinoma or hyperplastic lesion.

>> ï¼·ï½… confirmed all of samples were invasive cancer by our pathologists. We replaced them with new IHC image sets with the same magnitude of each cluster (A and B) for vimentin (Fig. 3C), CD8 (Fig 4d), and TIM3 (Fig 4e).

5) Carcinoma cells (or epithelial cells?) did not show positive immunoreactivity for vimentin, which show EMT. Since GBC cells did not show EMT, it may be inconsistent that the biological significance of miR125B1 and MIR1245A in terms of EMT was examined in GBC cells.

>> The increase in stroma and the increase in Vimentin-positive stromal cells around tumor cells means that fibroblast cells and stroma were induced by EMT of cancer cells. The decreased invasion ability by miR125B1 KO of cancer cells also biologically means the decreased activity of EMT pathway in cancer cell was observed. We explained this issue in Discussions.

“The increase of stroma and Vimentin-positive stromal cells indicates that fibroblast cells are induced by activated EMT pathway of cancer cells, and decreased invasion ability in cancer cells with miR125B1 knockout means biologically decreased activity of in EMT pathway in cancer cells.”

Minor

  • If the data on mutation analysis was partly included in authors’ previous study (J Hepatol 2018), it should be noted in the manuscript.

>>We add a comment about this issue in Methods.

“Clinical features of 39 patients with GBC and 8 HBDCs are shown in Table A1. DNAs of 38 GBCs samples were sequenced in our previous paper [21], however we called their mutations by different methods.”

2) Figure legends for Figure A4 and A5 are wrong. 

>> We correct this issue. Figure legends were swapped between A4 and A5. Thank you for your suggestion.

3) Figure A6(c): Matrigel invasion assay; there seems to be few differences between the Non-editting cells and miR125B1 KO cells 

>> We replaced them with enlarged pictures of the Matrigel invasion assays to make a difference clearer in Figure A6(c).

Reviewer 3 Report

I like to congratulate the authors for the manuscript and their holistic approach to the issue of molecular classification and tumor microenvironment characterization of gallbladder cancer by comprehensive genomic and transcriptomic analysis. The authors applied a large variety of methods (DNA/RNA seq, cell lines, immunohistochemistry, etc.) and all data is analyzed and presented well.

The manuscript has some drawbacks which have to be mentioned. First, the sample size is very small compared to other molecular classification cohorts especially from Japan. Secondly, the authors explore a variety of different subjects kind of "superficial". RNAseq - Focus on EMT and TGF signaling - Focus on immune genes and immune check-point molecules - Immunostaining for stroma markers and immune cells - MicroRNAs - TGF Signaling - Fusion genes (WGS/WES). Every single analysis is interesting and well presented but lacks an overall context or story which is continuously pursued through the entire manuscript, so the manuscript delivers interesting insights but does not result in a really meaningful conclusion. 

Minor points: Please add numbers at risk to the survival analysis; Minor language editing recommended (for example: "Invasion assays about cells line lacking miRNA125B1 showed may be target to treat GBC" within the simple summery). 

While not leading to a very meaningful conclusion, the analysis is solid and the large variety of methods impressive. For researchers interested in molecular classification of GBC, this manuscript might be very interesting and is worth publishing.  

Author Response

Reviewer 3

I like to congratulate the authors for the manuscript and their holistic approach to the issue of molecular classification and tumor microenvironment characterization of gallbladder cancer by comprehensive genomic and transcriptomic analysis. The authors applied a large variety of methods (DNA/RNA seq, cell lines, immunohistochemistry, etc.) and all data is analyzed and presented well.

The manuscript has some drawbacks which have to be mentioned. First, the sample size is very small compared to other molecular classification cohorts especially from Japan.

>> GBC is a rare type of cancer even in Japan. Usually it is diagnosed at a very advanced stage and resectable cases are also very rare. Hokkaido University Hospital is one of the highest volume centers for BTC in Japan, who resects about 3-4 GBC cases a year, and we collected 39 fresh-frozen surgical specimens for 15 years. Even TCGA had only one case of GBC, and this comprehensive data of 39 GBCs are valuable, we believe.

Secondly, the authors explore a variety of different subjects kind of "superficial". RNAseq - Focus on EMT and TGF signaling - Focus on immune genes and immune check-point molecules - Immunostaining for stroma markers and immune cells - MicroRNAs - TGF Signaling - Fusion genes (WGS/WES). Every single analysis is interesting and well presented but lacks an overall context or story which is continuously pursued through the entire manuscript, so the manuscript delivers interesting insights but does not result in a really meaningful conclusion. 

>> To clarify the conclusion of this study, we add conclusion sentences in Discussions and Abstract. One of the main results is TME, EMT, and TGF-beta pathway alterations in GBCs.

In Abstract, “This comprehensive analysis demonstrated that TME, EMT, and TGF-β pathway alterations are the main drivers of GBC and provides a new classification of GBCs that may be useful for therapeutic decision-making.”

In the last paragraph, “In summary, this gene clustering analysis on 36 GBCs reveals there are a sub-group with a poor prognosis, and it is characterized as TME-rich GBC which have the increased expression of EMT-related genes and TGF-beta pathway alterations. This demonstrated that TME, EMT, and TGF-β pathway alterations are the main drivers of GBC.”

Minor points: Please add numbers at risk to the survival analysis;

>> We add number of HR and 95% CI each survival curves in Table A2. In the legend of Figure 1, “Significant differences in the prognosis between Clusters A and B are observed (p = 0.01 and 0.005 by log-rank test, respectively, and HR: 6.6439, 95%CI [1.514-29.16] and HR: 3.623, 95%CI [1.226-10.71]).”

Minor language editing recommended (for example: "Invasion assays about cells line lacking miRNA125B1 showed may be target to treat GBC" within the simple summery). 

>> We corrected this sentences.

While not leading to a very meaningful conclusion, the analysis is solid and the large variety of methods impressive. For researchers interested in molecular classification of GBC, this manuscript might be very interesting and is worth publishing.  

>>Thank you for your positive evaluation of our study.
